# Bis(Disulfide)-Bridged Somatostatin-14 Analogs and Their [^111^In]In-Radioligands: Synthesis and Preclinical Profile

**DOI:** 10.3390/ijms25031921

**Published:** 2024-02-05

**Authors:** Aikaterini Tatsi, Theodosia Maina, Beatrice Waser, Eric P. Krenning, Marion de Jong, Jean Claude Reubi, Paul Cordopatis, Berthold A. Nock

**Affiliations:** 1Molecular Radiopharmacy, INRaSTES, NCSR “Demokritos”, GR-15341 Athens, Greece or ktatsi@upatras.gr (A.T.); nock_berthold.a@hotmail.com (B.A.N.); 2Department of Pharmacy, University of Patras, GR-26500 Patras, Greece; 3Institute of Pathology, University of Berne, CH-3010 Berne, Switzerland; waserpatho@rubigen.ch (B.W.); jean.reubi@pathology.unibe.ch (J.C.R.); 4Cyclotron Rotterdam BV, Erasmus MC, 3015 Rotterdam, The Netherlands; erickrenning@gmail.com; 5Department of Radiology and Nuclear Medicine, Erasmus MC, 3015 Rotterdam, The Netherlands

**Keywords:** DOTA-conjugated somatostatin, [^111^In]In-radioligand, bicyclic somatostatin, metabolic stability, tumor targeting, theranostics

## Abstract

The overexpression of one or more somatostatin receptors (SST_1–5_R) in human tumors has provided an opportunity for diagnosis and therapy with somatostatin-like radionuclide carriers. The application of “pansomatostatin” analogs is expected to broaden the clinical indications and upgrade the diagnostic/therapeutic efficacy of currently applied SST_2_R-prefering radioligands. In pursuit of this goal, we now introduce two bicyclic somatostatin-14 (SS14) analogs, AT5S (DOTA-Ala^1^-Gly^2^-c[Cys^3^-Lys^4^-Asn^5^-c[Cys^6^-Phe^7^-DTrp^8^-Lys^9^-Thr^10^-Cys^11^]-Thr^12^-Ser^13^-Cys^14^]) and AT6S (DOTA-Ala^1^-Gly^2^-c[Cys^3^-Lys^4^-c[Cys^5^-Phe^6^-Phe^7^-DTrp^8^-Lys^9^-Thr^10^-Phe^11^-Cys^12^]-Ser^13^-Cys^14^]), suitable for labeling with trivalent radiometals and designed to sustain in vivo degradation. Both AT5S and AT6S and the respective [^111^In]In-AT5S and [^111^In]In-AT6S were evaluated in a series of in vitro assays, while radioligand stability and biodistribution were studied in mice. The 8/12-mer bicyclic AT6S showed expanded affinity for all SST_1–5_R and agonistic properties at the SST_2_R, whereas AT5S lost all affinity to SST_1–5_R. Both [^111^In]In-AT5S and [^111^In]In-AT6S remained stable in the peripheral blood of mice, while [^111^In]In-AT6S displayed low, but specific uptake in AR4-2J tumors and higher uptake in HEK293-SST_3_R tumors in mice. In summary, high radioligand stability was acquired by the two disulfide bridges introduced into the SS14 motif, but only the 8/12-mer ring AT6S retained a pansomatostatin profile. In consequence, [^111^In]In-AT6S targeted SST_2_R-/SST_3_R-positive xenografts in mice. These results call for further research on pansomatostatin-like radioligands for cancer theranostics.

## 1. Introduction

The clinical advent of OctreoScan^®^ ([^111^In]In-DTPA-octreotide; DTPA, diethylenetriaminepentaacetic acid; octreotide, H-DPhe^1^-c[Cys^2^-Phe^3^-DTrp^4^-Lys^5^-Thr^6^-Cys^7^]-Thr^8^-ol) in the diagnostic imaging of neuroendocrine tumors (NET) with high expression of the somatostatin subtype 2 receptor (SST_2_R) in the early 1990s has been a game changer in nuclear medicine [1,2,3,4]. In the following years, a plethora of cyclic octapeptide analogs, based on octreotide and carrying macrocyclic chelators to stably bind medically relevant radiometals, were developed and evaluated in preclinical models and in patients. As a result, [^68^Ga]Ga/[^177^Lu]Lu-DOTA-TATE (DOTA, 1,4,7,10-tetraazacyclododecane-1,4,7,10-tetraacetic acid; TATE, [Tyr^3^,Thr^8^]octreotide) and [^68^Ga]Ga/[^177^Lu]Lu-DOTA-TOC (TOC, [Tyr^3^]octreotide) are currently approved radiopharmaceuticals broadly used in the management of NET patients [5,6,7,8,9,10]. Along these lines, diagnostic imaging is applied to identify patients with SST_2_R-positive lesions and hence eligible for radionuclide therapy. Furthermore, imaging is essential for dosimetry and therapy planning, as well as for monitoring therapy outcomes according to a personalized theranostic approach [11].

Although the SST_2_R is the predominant receptor subtype in most NET lesions, there are many instances whereby it is co-expressed with other somatostatin receptor subtypes (SST_1–5_R) [12,13,14,15]. For example, the SST_2_R is frequently found together with SST_5_R in growth hormone (GH)-secreting pituitary adenomas, or in various combinations, e.g., with SST_1_R, in gastroenteropancreatic (GEP)-NETs [16,17,18]. Furthermore, one or more SST_1–5_R may be expressed in tumors devoid of SST_2_R [12,15,16,17,18,19,20,21,22], such as in exocrine pancreatic ductal adenocarcinoma [19,20], or in primary androgen-dependent prostate cancer [21,22]. Based on these findings, it is rational to assume that radioligands displaying high affinity to all five SST_1–5_R can be used in broader clinical indications and show enriched diagnostic accuracy or therapeutic efficacy compared to SST_2_R-prefering analogs. It should be noted that octreotide, per se, shows affinity to both SST_2_R and SST_5_R, whereas some of its analogs display broader SST_1–5_R profiles. For example, DOTA-NOC (DOTA-1-Nal^3^-octreotide; 1-Nal, 1-naphtyl-Ala), DOTA-NOCATE (DOTA-1-Nal^3^-Thr^8^-octreotide), DOTA-BOC (DOTA-1-BzThi^3^-octreotide; 1-BzThi, benzothienyl-Ala), and DOTA-BOCATE (DOTA-1-BzThi^3^-Thr^8^-octreotide) exhibited extended affinity to SST_2_R, SST_3_R, and SST_5_R, while their radioligands preserved the metabolic stability of OctreoScan^®^ [23,24,25,26]. To promote somatostatin-based therapy, two more multi-somatostatin receptor analogs were introduced next, the 6-mer ring SOM230 (c[Hyp(Unk)-Phg-DTrp-Lys-Tyr(Bn)-Phe]; Hyp(Unk), 2-aminoethyl-carbamoyl-oxy-Pro; Phg, phenylGly; Bn, benzyl) with high affinity to SST_1–3,5_R [27] and the 8-mer ring KE108 (Tyr^0^-c[DDab-Arg-Phe-Phe-DTrp-Lys-Thr-Phe]; Dab, diaminobutyric acid) with a pansomatostatin affinity profile [28]. However, the inability of SOM230 or KE108 to induce SST_2_R internalization turned out to severely reduce the uptake of their respective radioligands in SST_2_R-positive cells and tumors. Hence their diagnostic/therapeutic value was compromised [29,30,31].

Joining this effort, we have previously reported on a series of [^111^In]In-radioligands obtained by the covalent coupling of DOTA to the N-terminal of native somatostatin 14 (SS14) or somatostatin 28 (SS28) analogs [32,33]. As expected, the resulting compounds showed a pansomatostatin affinity profile, but uptake in implanted SST_2_R/SST_3_R/ SST_5_R-positive tumors in mice was found dependent on in vivo stability. Indeed, SS14 and, to a lesser extent, SS28 were rapidly degraded in vivo by the ectoenzyme neprilysin (NEP) [34,35]. By co-injection of the potent NEP-inhibitor phosphoramidon (PA) with [^111^In]In-AT1S (AT1S, DOTA-Ala^1^-Gly^2^-c[Cys^3^-Lys^4^-Asn^5^-Phe^6^-Phe^7^-Trp^8^-Lys^9^-Thr^10^-Phe^11^-Thr^12^-Ser^13^-Cys^14^-OH]; Figure 1a), we were able to induce marked radioligand stabilization (from <2% intact in controls to 86% intact in PA-mice at 5 min post-injection, pi) and significant uptake increases in AR4-2J xenografts (calculated as intact activity per g of tissue (%IA/g)—from 1%IA/g to 14%IA/g at 4 h pi), confirming the involvement of NEP in [^111^In]In-AT1S catabolism [36]. Furthermore, we investigated a series of AT1S analogs with decreasing ring-size (from 12-mer to 6-mer), while keeping the total number of amino acids to 14, in search of pansomatostatin and NEP-resistant radioligands [37]. Ring-size and stereochemistry turned out to have a profound effect on the SST_1–5_R affinity profile, SST_2_R-internalization, and in vivo stability. Interestingly, the 6-mer ring AT3S (DOTA-Ala^1^-Gly^2^-Nle^3^-Lys^4^-Asn^5^-c[Cys^6^-Tyr^7^-DTrp^8^-Lys^9^-Thr^10^-Cys^11^]-Thr^12^-Ser^13^-Gly^14^) lost all affinity to SST_1–5_R but [^111^In]In-AT3S remained >90% intact in the blood of mice. In contrast, the 8-mer ring AT4S (DOTA-Ala^1^-Gly^2^-c[Cys^3^-Lys^4^-Asn^5^-c[Cys^6^-Phe^7^-DTrp^8^-Lys^9^-Thr^10^-Cys^11^]-Thr^12^-Ser^13^-Cys^14^]) showed a pansomatostatin affinity profile and [^111^In]In-AT4S displayed marked stability improvements (>68% intact) compared with [^111^In]In-AT1S.

Intrigued by these findings, we decided to study two additional bicyclic analogs with either a double 6/12-mer ring (AT5S, DOTA-Ala^1^-Gly^2^-c[Cys^3^-Lys^4^-Asn^5^-c[Cys^6^-Phe^7^-DTrp^8^-Lys^9^-Thr^10^-Cys^11^]-Thr^12^-Ser^13^-Cys^14^]) or a double 8/12-mer ring (AT6S, DOTA-Ala^1^-Gly^2^-c[Cys^3^-Lys^4^-c[Cys^5^-Phe^6^-Phe^7^-DTrp^8^-Lys^9^-Thr^10^-Phe^11^-Cys^12^]-Ser^13^-Cys^14^]) (Figure 1). By carefully monitoring their biological responses in the same experimental setting, we aimed to further explore the somatostatin system. Important new data may be smartly exploited to better design pansomatostatin radioligands for successful use in cancer theranostics.

## 2. Results

### 2.1. Synthesis of Bicyclic Peptide-Conjugates

#### 2.1.1. Synthesis of AT5S

For the synthesis of AT5S, the linear protected sequence DOTA(tris(*^t^*Bu))-Ala^1^-Gly^2^-Cys(Acm)^3^-Lys(Boc)^4^-Asn(Trt)^5^-Cys(Trt)^6^-Phe^7^-DTrp(Boc)^8^-Lys(Boc)^9^-Thr(*^t^*Bu)^10^-Cys(Trt)^11^-Thr(*^t^*Bu)^12^-Ser(*^t^*Bu)^13^-Cys(Acm)^14^- was built on the H-Cys(Acm)-2-Cl-Trt resin adopting Fmoc/*^t^*Bu methodologies (Figure 2a; and detailed in Section 4.1 and Section 4.2 in Section 4). Couplings were successful, but the DOTA(tris(*^t^*Bu)) chelator precursor required two rounds to full attachment. Release from the resin with concomitant deprotection of lateral groups, except for Acm, was achieved by a TFA/TΙS/EDT/anisole/H2O (93:3:2:1:1) solution. Next, the first disulfide bond formation between Cys^6^ and Cys^11^ and the generation of the respective 6-mer ring followed, using a mixture of 0.01 M phosphate, pH 7.5, and 1% DMSO. This step turned out to be slow (27 h) as monitored by the Ellman test and reverse-phase high performance liquid chromatography (RP-HPLC). The product was purified by RP-HPLC twice and its formation was confirmed by electrospray-ionization mass-spectroscopy (ESI-MS). The second oxidation conducted by I_2_ in aqueous acetic acid was completed rapidly (25 min) under concomitant Acm removal and formation of the second disulfide Cys^3^-Cys^14^ bond (12-mer ring) [38]. The bicyclic peptide-conjugate was isolated by RP-HPLC to 94% purity and the ESI-MS result was in agreement with the expected structure (Table 1) in a 27% yield; the latter was lower from the one reported for other monocyclic ATXS products [37]. This was attributed to the high number of by-products from the first oxidation step, necessitating repeated purification by RP-HPLC prior to moving further to the second oxidation and the closure of the 12-mer ring.

#### 2.1.2. Synthesis of AT6S

A similar process was followed for the synthesis of AT6S. This time, the protected linear sequence DOTA(tris(*^t^*Bu))-Ala^1^-Gly^2^-Cys(Acm)^3^-Lys(Boc)^4^-Cys(Trt)^5^-Phe^6^-Phe^7^-DTrp(Boc)^8^-Lys(Boc)^9^-Thr(*^t^*Bu)^10^-Phe^11^-Cys(Trt)^12^-Ser(*^t^*Bu)^13^-Cys(Acm)^14^ was built on the same resin (Figure 2b; and detailed in Section 4.1 and Section 4.2 in Section 4), but all coupling reactions were straightforward, not requiring repetition. The two disulfide bonds, Cys^5^-Cys^12^ (8-mer ring) and Cys^3^-Cys^14^ (12-mer ring), were consecutively formed following the same protocol used for AT5S. Again, the first cyclization using DMSO required longer reaction times (24 h), careful monitoring, and isolation of the product by RP-HPLC prior to starting the second cyclization step, which lasted only 25 min. The final compound was isolated by RP-HPLC to >93% purity in an overall yield of 27% and the ESI-MS result was consistent with the expected structure (Table 1 and Appendix A).

### 2.2. Radiochemistry

#### Radiolabeling of AT5S and AT6S with In-111-Quality Control

Labeling of the two bioconjugates with In-111 was achieved by heating at 95 °C for 20 min at pH 4.6 to facilitate the full incorporation of the radiometal without formation of undesirable hydroxides. Radiochemical purities of >94% were achieved at a molar activity of 3.7–7.4 MBq/nmol. As verified by RP-HPLC analysis, a single radiopeptide species was obtained each time and therefore, the forming [^111^In]In-AT5S and [^111^In]In-AT6S radioligands were used without further purification in all subsequent biological tests. Moreover, the preservation of the radiochemical purity of both [^111^In]In-AT5S and [^111^In]In-AT5S was confirmed prior to and at the end of the assays by RP-HPLC.

### 2.3. In Vitro Studies

#### 2.3.1. Affinity Profile of AT5S and AT6S to the Five Human SST_1–5_R

The binding affinities of AT5S and AT6S to each of the five human SST_1–5_R (hSST_1–5_R) were evaluated in CHO (hSST_1_R), CCL39 (hSST_2_R, hSST_3_R, or hSST_4_R), or HEK293 (hSST_5_R) cells transfected to stably express one of the above subtypes. Competition binding assays against the pansomatostatin radioligand [^125^I][Leu^8^,DTrp^22^,Tyr^25^]SS28 ([^125^I]I-[LTT]SS28) were performed in suitably prepared frozen slices of the cells. Receptor autoradiography methods were applied to determine half-maximum inhibitory constants (IC_50_), using SS14 and AT2S as reference compounds [32,37]. The results summarized in Table 2 represent mean IC_50_ values ± SEM (standard error of the mean) in nM.

As shown in Table 2, AT5S with a 6- and a 12-member ring, lost affinity binding to all five hSST_1–5_R, unlike AT6S with an 8- and a 12-member ring, which exhibited a clear pansomatostatin affinity profile. Compared with AT2S [32,37], the bicyclic AT6S showed comparable affinities for the hSST_1_R and hSST_4_R, but lower affinities for the hSST_2_R (<4 fold), hSST_3_R (≈4 fold) and hSST_5_R (<2 fold). Compared with the native hormone, a more pronounced drop in binding affinities was observed for all of the somatostatin receptors, hSST_1_R (<6 fold), hSST_2_R (9 fold), hSST_3_R (≈3 fold), hSST_4_R (<3 fold) and hSST_5_R (<6 fold).

#### 2.3.2. Ligand-Induced Internalization of the hSST_2_R

An immunofluorescence microscopy-based internalization assay was performed in HEK293 cells transfected to stably express the hSST_2_R tagged with the T7-epitope (HEK293-SST_2_R) to determine the agonistic/antagonistic properties of AT5S and AT6S, using native SS14, TOC, and AT1S as reference agonists [32]. As shown in Figure 3, in the absence of a hSST_2_R-ligand, the receptor remains on the cell membrane of the cells (a), but the known agonists SS14 (at 10 nM, (b)), TOC (at 10 nM—(c); 1 µM—(d)), and AT1S (at 1 µM, (e)) were clearly able to trigger the internalization of hSST_2_R. On the other hand, the 6/12-member ring AT5S failed to trigger the hSST_2_R internalization at 1 µM (f) or even at 10 µM (g). It furthermore failed to prevent the hSST_2_R internalization induced by 10 nM SS14 in the same cells, even at a concentration of 10 µM (h). Therefore, it behaves neither as an agonist nor as an antagonist in this assay, a finding attributed to its poor affinity for the hSST_2_R. In contrast, the 8/12-member ring bicyclic AT6S, exhibiting a single-digit binding affinity for the hSST_2_R, behaved as a potent agonist triggering the internalization of the receptor at 1 µM (i).

#### 2.3.3. Internalization of [^111^In]In-AT6S in AR4-2J and HEK293-hSST_3_R Cells

The internalization of [^111^In]In-AT6S was studied in AR4-2J cells that endogenously express the rat SST_2_R (rSST_2_R) [39,40] and in HEK293 cells transfected to stably express the hSST_3_R (HEK293-hSST_3_R). The [^111^In]In-AT5S was not included in this study, in view of its lack of affinity for any of the hSST_1–5_R, as well as its lack of agonist/antagonist behavior at the hSST_2_R during the immunofluorescence microscopy-based analysis. For [^111^In]In-AT6S, assays were conducted by a 1 h incubation at 37 °C at 1–2 nM without or in the presence of 1 µM blocker to determine non-specific internalization (TATE for rSST_2_R blockade in AR4-2J cells) and KE108 (for hSST_3_R blockade in HEK293-hSST_3_R cells) [28]. The results are summarized in Figure 4, revealing low internalization (0.63 ± 0.10%) and overall uptake (1.17 ± 0.39%) of the radioligand in the AR4-2J cells, with these values turning out to be clearly higher in the HEK293-hSST_3_R cells (2.35 ± 0.63% and 4.76 ± 0.67%, respectively).

The percentage of specific internalization of [^111^In]In-AT6S was found to be lower when compared with either [^111^In]In-AT1S (AR4-2J: 1.30 ± 0.37% and HEK293-hSST_3_R: 4.43 ± 1.61%) or [^111^In]In-AT2S (AR4-2J: 3.72 ± 0.83% and HEK293-hSST_3_R: 6.08 ± 1.62%) [32], revealing the unfavorable influences of the double 8/12-mer rings on internalization rates.

### 2.4. Animal Studies

#### 2.4.1. Metabolic Stability of [^111^In]In-AT5S and [^111^In]In-AT6S in Mice

The metabolic stability of [^111^In]In-AT5S and [^111^In]In-AT6S after their entry into the circulation of mice was studied by the analysis of blood samples collected 5 min post-injection (pi) by radio-HPLC. Representative radiochromatograms are included in Figure 5, revealing the high metabolic stability of the bicyclic analogs.

An impressive improvement of stability is observed for both bicyclic analogs compared with those previously reported for monocyclic [^111^In]In-AT1S (2.2% intact) and [^111^In]In-AT2S (6.5% intact) [32]. The rapid degradation of the latter two monocyclic radioligands in the peripheral blood of mice could be assigned to the rapid proteolytic action of neprilysin (NEP) [34,36]. It should be noted that the 6-member ring monocyclic [^111^In]In-AT3S (AT3S, DOTA-Ala^1^-Gly^2^-Nle^3^-Lys^4^-Asn^5^-c[Cys^6^-Tyr^7^-DTrp^8^-Lys^9^-Thr^10^-Cys^11^]-Thr^12^-Ser^13^-Gly^14^-OH) also displayed high metabolic stability (>96% intact at 5 min pi) [37]. On the other hand, a significant improvement was achieved in the metabolic stability of [^111^In]In-AT6S (>98% intact) vs. the corresponding monocyclic 8-member ring [^111^In]In-AT4S (AT3S, DOTA-Ala^1^-Gly^2^-Nle^3^-Lys^4^-c[Cys^5^-Phe^6^-Phe^7^-DTrp^8^-Lys^9^-Thr^10^-Phe^11^-Cys^12^]-Ser^13^-Gly^14^-OH; >68% intact) [37], as a result of the second 12-member ring closure.

#### 2.4.2. Biodistribution of [^111^In]In-AT6S in Mice

The bicycle 6/12-member ring [^111^In]In-AT5S was not included in the biodistribution studies, due to its inability to interact with any of the five hSST_1–5_R. The biodistribution of [^111^In]In-AT6S was assessed in rSST_2_R-positive AR4-2J tumors raised in severe combined immunodeficiency disease (SCID) mice at 4 and 24 h pi and during in vivo SST_2_R-blockade at 4 h pi induced by co-injection of excess TATE [32]. In addition, [^111^In]In-AT6S was studied in HEK293-hSST_3_R xenograft-bearing SCID mice at 4 h pi and during in vivo SST_3_R-blockade at 4 h pi by co-injection of excess KE108 [28]. The results, calculated as percent injected activity per gram tissue (%IA/g), are summarized in Table 3 and represent mean values ± sd (n = 4). Moreover, the comparative uptake of [^111^In]In-AT6S and [^111^In]In-AT2S for the AR4-2J and the HEK293-hSST_3_R tumors, as well as in mice kidneys at 4 h pi [32] is included in Figure 6.

The radioligand exhibited low levels in the blood at 4 h pi and showed visible uptake in the liver, spleen, intestines, and especially in the kidneys, but uptake declined further at 24 h pi. Uptake in the AR4-2J tumors was low at 4 h pi (1.9 ± 0.1%IA/g) and significantly dropped with the co-injection of excess TATE (0.8 ± 0.05%IA/g; *p* < 0.001). The AR4-2J tumor values of [^111^In]In-AT6S declined at 24 h pi (0.8 ± 0.1%IA/g). In the animals bearing the HEK293-hSST_3_R xenografts, a similar pattern was observed in healthy organs and tissues at 4 h pi, but [^111^In]In-AT6S achieved a markedly higher uptake in the HEK293-hSST_3_R tumors (3.7 ± 0.4%IA/g), in line with its higher uptake in the HEK293-hSST_3_R cells. Tumor uptake could be significantly reduced during hSST_3_R-blockade by co-injection of excess KE108 (0.3 ± 0.05%IA/g; *p* < 0.001), confirming a hSST_3_R-mediated process. It is interesting to observe the drop of renal uptake in both groups of blocked animals by co-injection of either TATE (from 86.7 ± 10.9%IA/g to 49.5 ± 10.5%IA/g; *p* < 0.001) or KE108 (from 61.1 ± 10.6%IA/g to 26.8 ± 5.4%IA/g; *p* < 0.001). Both peptide analogs carry pendant primary amines (Lys^5^ in TATE; DDab^1^/Lys^6^ in KE108), which may partially saturate the cubilin/megalin system in the endocytic apparatus of the renal proximal tubule of the kidneys [41,42], thereby facilitating the excretion of radioactivity into urine.

Compared with the quickly biodegradable [^111^In]In-AT2S (≈6% intact at 5 min pi in the peripheral blood of mice) [32], [^111^In]In-AT6S displayed similar uptake in the AR4-2J tumors but a notably higher uptake in the HEK293-hSST_3_R xenografts (Figure 6a,b, respectively). Of particular interest are the elevated renal values of [^111^In]In-AT6S compared to [^111^In]In-AT2S, tentatively attributed to their different metabolic fates and further processing of [^111^In]In-AT2S-derived radiometabolites (Figure 6c).

## 3. Discussion

Previous efforts to develop metabolically robust pansomatostatin-like radioligands for cancer theranostics were driven by the co-expression of SST_1–5_Rs in different combinations in NET and other tumor lesions [8,13,14,15,16,17,18,27,28,31,43]. The accomplishment of this goal has been hitherto impeded by two major challenges. On one hand, SS14 and SS28 and their radiolabeled pansomatostatin-like analogs, such as [^111^In]In-AT1S and [^111^In]In-AT2S, were shown to undergo fast enzymatic degradation after entering the circulation [32,34,36]. On the other hand, less biodegradable synthetic analogs of smaller ring size ended up with partial or total loss of SST_1–5_R affinity. In addition, such analogs have been often linked with the absence of SST_2_R-internalization as well as unfavorable pharmacokinetics in mice, compromising their clinical applicability [29,30,31,33,37,44]. It should be stressed that the currently applied octreotide-based radioligands, like [^68^Ga]Ga/[^177^Lu]Lu-DOTA-TOC and [^68^Ga]Ga/[^177^Lu]Lu-DOTA-TATE, are actually SST_2_R-preferring [5,6,8,9,11]. Their clinical success relies on the 6-member ring octapeptide structure leading to high metabolic stability and to enhanced internalization in cancer cells [6,8,31]. These features, in combination with the prevailing SST_2_R-expression in NETs and other human tumors, have synergistically contributed to the high diagnostic accuracy and therapeutic efficacy achieved in patients [5,7,15].

In contrast to the above, we have observed that 6-member ring analogs containing a total of 14 amino acids, like AT3S, unexpectedly lost affinity to all SST_1–5_Rs, while the respective [^111^In]In-AT3S remained stable in mouse circulation [37]. Furthermore, the 8-member ring 14-peptide analog, AT4S, showed a pansomatostatin affinity profile and the corresponding [^111^In]In-AT4S was notably more stable compared to the 12-mer ring analogs [^111^In]In-AT1S and [^111^In]In-AT2S [32,37]. In a continuation of this work, we have now synthesized two bicyclic 6/12-mer and 8/12-mer ring 14-peptide analogs by introducing two disulfide bridges in the SS14 scaffold, namely, Cys^6^-Cys^11^/Cys^3^-Cys^14^ for AT5S and Cys^5^-Cys^12^/Cys^3^-Cys^14^ for AT6S (Figure 1). In this way, the issue of metabolic stability is vigorously addressed for both. Concurrently, the effects of enhanced molecule rigidity on SST_1–5_R affinity, internalization, and tumor uptake could be compared with their respective monocyclic counterparts (AT3S and AT4S) [37], as well as with the AT1S/AT2S prototypes [32].

The first results on SST_1–5_R affinity, obtained by receptor autoradiography on cells expressing one of the five receptors are summarized in Table 1. The introduction of the second Cys^3^-Cys^14^ disulfide bridge in AT5S failed to improve the affinity to any of the five SST_1–5_R when compared with monocyclic AT3S [37]. However, the same intervention (introduction of a second Cys^3^-Cys^14^ disulfide bridge) in AT6S led to affinity enhancement in all five receptor subtypes with the exception of SST_4_R, compared with the monocyclic 8-mer ring AT4S [37], revealing the positive influence of the increased rigidity of the bicyclic molecule. The unexpected lack of SST_1–5_R affinity of AT3S and AT5S is truly intriguing, in view of the high SST_2_R-affinity of related 6-mer monocyclic octapeptide analogs, such as octreotide, having identical 6 amino acid rings. Dedicated studies using NMR methodology may reveal conformational changes between 6-mer monocyclic and 6/12-mer bicyclic tetradecapeptide analogs compared to their octapeptide counterparts, causing this discrepancy in affinity, especially on the SST_2_R. The agonist/antagonist properties of AT5S and AT6S were studied next by an immunofluorescence microscopy-based internalization assay in HEK293-SST_2_R cells and the results are summarized in Figure 3. As expected by its poor SST_2_R-affinity, AT5S failed not only to trigger SST_2_R-internalization in concentrations up to 10 µM, but also to inhibit the SS14-induced internalization of the receptor even in high molar excess (10 µM to 10 nM). Hence, AT5S behaved neither as an agonist nor as an antagonist at the SST_2_R. On the other hand, the 8/12-mer bicyclic AT6S, similarly to its monocyclic AT4S counterpart [37], behaved as a strong agonist in the same assay by stimulating the SST_2_R-internalization at 1 µM. Lastly, the internalization of [^111^In]In-AT6S was investigated in AR4-2J and HEK293-SST_3_R cells (Figure 4). Internalization in AR4-2J cells was comparable to that of the monocyclic 8-mer ring [^111^In]In-AT4S [37], but clearly inferior to the internalization of the monocyclic 12-mer ring [^111^In]In-AT1S and [^111^In]In-AT2S [32]. Interestingly, the internalization of [^111^In]In-AT6S was more pronounced in the HEK293-SST_3_R cells. A similar higher internalization in HEK293-SST_3_R cells than in HEK293-SST_2_R cells was reported for [^111^In]In-DOTA-LTT-SS28 [33]. The latter internalized much more efficiently compared to [^111^In]In-AT6S in both cell lines, most probably due to the higher affinity of DOTA-LTT-SS28 and [^nat^In]In-DOTA-LTT-SS28 for SST_2_R and SST_3_R [33].

The bicyclic 6/12-mer [^111^In]In-AT5S and 8/12-mer [^111^In]In-AT6S showed >96% stability in the peripheral blood of mice, in contrast to the monocyclic 12-mer ring [32]. The rapid cleavage of peptide bonds in the SS14 motif could be previously assigned to NEP by monitoring the formation of (radio)metabolites without or in the presence of potent NEP-inhibitors, like PA and thiorphan [34,35,36,45]. It is interesting to note that increased stability was already observed for the respective 6-mer ring [^111^In]In-AT3S (96%) and 8-mer ring [^111^In]In-AT4S (>68%) 14-peptide analogs. Furthermore, the DCys^5^-substituted 8-mer ring [^111^In]In-AT9S showing moderate affinities to all SST_1–5_R was >92% stable in the same assay. These findings confirm previous observations that molecule rigidity imposed by the two disulfide bridges, along with ring size and conformation play a critical role in the in vivo stability of SS14-based radioligands.

The biodistribution of pansomatostatin-like and in vivo robust [^111^In]In-AT6S was studied in mice bearing either AR4-2J (rSST_2_R-positive) or HEK293-hSST_3_R xenografts with the aim to evaluate its suitability for clinical use (Table 3 and Figure 6). Compared with the quickly biodegradable but more SST_2_R-affine [^111^In]In-AT2S, [^111^In]In-AT6S showed similar levels of uptake in AR4-2J tumors, but superior uptake in HEK293-hSST_3_R tumors, emphasizing the importance of metabolic stability for good tumor localization [32,37]. This assumption was further supported by the high values attained by the fairly stable and highly SST_1–5_R-affine [^111^In]In-DOTA-LTT-SS28 in the same tumor models [33]. Unluckily, by increasing the stability of SS14-based analogs, an unfavorably increasing kidney accumulation was most often observed, which was found to be overwhelming in the case of [^111^In]In-DOTA-LTT-SS28 [32,33,36,37]. Administration of positive amino acid solutions alone or together with plasma expanders, such as Gelofusine, have been previously proposed to mitigate this problem [41,42]. It is interesting to compare our findings with another class of bicyclic somatostatin-like peptides comprising an octreotide ring and a head-to-tail-coupled Arg-Dab(DOTA) 8-mer ring cycle, such as AM3 (DOTA-Tyr-c{[Dab-Arg-c[Cys-Phe-DTrp-Lys-Thr-Cys]}) [46]. These analogs were SST_2_R/SST_3_R-preferring, having reduced affinities for SST_1_R and SST_5_R. However, their [^68^Ga]Ga/^177^Lu]Lu-radioligands were likewise metabolically robust. They displayed superior internalization in SST_2_R/SST_3_R-positive cells and hence in respective tumors xenografted in mice. Kidney values were moderate and were reduced by half by pre-treatment of mice with a Lys solution [46].

## 4. Materials and Methods

### 4.1. Materials and Instrumentation

#### 4.1.1. Chemicals

All chemicals were reagent grade and used without further purification. The protected chelator 2-(4,7,10-tris(2-tert-butoxy-2-oxoethyl)-1,4,7,10-tetraazacyclo-dodecan-1-yl)acetic acid (DOTA-tris(tert-butyl) (*^t^*Bu) ester;) was supplied by CheMatech (Dijon, France). The L-amino acid precursors (9-fluorenylmethyloxycarbonyl (Fmoc))-Ala-OH, Fmoc-Gly-OH, Fmoc-Cys(triphenylmethyl) (Trt)-OH), Fmoc-Lys(tert-butyloxycarbonyl) (Boc)-OH, Fmoc-Asn(Trt)-OH, Fmoc-Phe-OH, Fmoc-Thr(*^t^*Bu)-OH, Fmoc-Ser(*^t^*Bu)-OH, the D-amino acid precursor Fmoc-DTrp(Boc)-OH, and H-L-Cys(S-acetamidomethyl) (Acm)-2-chlorotrityl resin (substitution 0.40 mmol/g) that was used in solid-phase peptide synthesis (SPPS) were purchased from CBL (Patras, Greece). The L-amino acid precursor Fmoc-Cys(Acm)-OH was purchased from Novabiochem (Darmstadt, Germany). For the coupling of the Fmoc-amino acids and the protected chelator DOTA-tris(*^t^*Bu)ester, the activating agents 1-hydroxybenzotriazol (HOBt) (CBL, Patras, Greece) and N,N′-diisopropylcarbodiimide (DIC) (Sigma Aldrich, Steinheim, Germany) were employed. The In-111 used for labeling was purchased from Mallinckrodt Medical B.V. (Petten, The Netherlands) in the form of [^111^In]InCl_3_ in a solution of 50 mM HCl at a 370 MBq/mL activity concentration on calibration date.

#### 4.1.2. Analysis-Radiochemistry

The peptide products were purified by semipreparative HPLC using a Mod.10 ÄKTA system from Amersham Biosciences (Piscataway, NJ, USA). A Supelcosil C18 (5 μm, 8 mm × 250 mm) column from Sigma Aldrich (St. Louis, MO, USA) was eluted at a flow rate of 2 mL/min with a linear gradient from 90%A/10%B to 40%A/60% B in 50 min, whereby A = 0.1% aqueous trifluoroacetic acid (TFA) and B = 0.1% TFA in MeCN. Formation of expected products was confirmed by ESI-MS using a Micromass-Platform LC instrument (Waters Micromass Technologies, Milford, MA, USA). The purity of the final products was assessed by a Waters Chromatograph (Waters, Vienna, Austria) with a 600E multi-solvent delivery system coupled to a Waters 2998 photodiode array UV detector. Analyses were performed System 1: on an XBridge^TM^ Shield RP18 cartridge column (5 μm, 4.6 mm × 150 mm; Waters, Eschborn, Germany) eluted at a 1 mL/min flow rate with a linear gradient from 80%A/20%B to 60%A/40%B in 40 min, and System 2: RP-HPLC on a Symmetry C18 analytical column (3.5 μm, 4.6 mm × 75 mm; Waters, Micromass Technologies, Milford, MA, USA) eluted at a 1 mL/min flow rate with a linear gradient from 90%A/10%B to 50%A/50%B in 20 min, whereby A = 0.1% aqueous TFA, B = MeCN. Data processing and chromatographic control were conducted using Empower 2 software (Waters, Eschborn, Germany). Radiochemical HPLC analyses were performed on the same Waters Chromatograph coupled to a Gabi γ-detector (Raytest, RSM Analytische Instrumente GmbH, Straubenhardt, Germany). Radioactivity measurements were conducted in an automated well-type γ-counter (an automated multi-sample well-type instrument with a NaI(Tl) 3″ crystal, Canberra Packard Cobra^TM^ Quantum U5003/1, Auto-Gamma^®^ counting system; Little Rock, AR/USA).

### 4.2. Synthesis of the Bicyclic DOTA-SS14 Peptide Conjugates

#### 4.2.1. Peptide-Chain Assembly and Chelator-Coupling

The orthogonally protected sequences: DOTA(tris(*^t^*Bu)ester)-Ala^1^-Gly^2^-Cys(Acm)^3^-Lys(Boc)^4^-Asn(Trt)^5^-Cys(Trt)^6^-Phe^7^-DTrp(Boc)^8^-Lys(Boc)^9^-Thr(*^t^*Bu)^10^-Cys(Trt)^11^-Thr(*^t^*Bu)^12^-Ser(*^t^*Bu)^13^-Cys(Acm)^14^ (AT5S) and DOTA(tris(*^t^*Bu)ester)-Ala^1^-Gly^2^-Cys(Acm)^3^-Lys(Boc)^4^-Cys(Trt)^5^-Phe^6^-Phe^7^-DTrp(Boc)^8^-Lys(Boc)^9^-Thr(*^t^*Bu)^10^-Phe^11^-Cys(Trt)^12^-Ser(*^t^*Bu)^13^-Cys(Acm)^14^ (AT6S) were assembled on an H-L-Cys(Acm)-2-chlorotrityl resin (250 mg with a 0.4 mmol/g loading) applying Fmoc/*^t^*Bu methodology. The elongation of the protected peptide chain was performed step by step for each residue in 3 molar excess (0.3 mmol), HOBt (69 mg, 0.45 mmol) and DIC (51.1 µL, 0.33 mmol). After coupling of the terminal Fmoc-Ala-OH, the Fmoc group was removed and the peptide-loaded resin, either Ala^1^-Gly^2^-Cys(Acm)^3^-Lys(Boc)^4^-Asn(Trt)^5^-Cys(Trt)^6^-Phe^7^-DTrp(Boc)^8^-Lys(Boc)^9^-Thr(*^t^*Bu)^10^-Cys(Trt)^11^-Thr(*^t^*Bu)^12^-Ser(*^t^*Bu)^13^-Cys(Acm)^14^-O-Cl-Trt-R (AT5S) or Ala^1^-Gly^2^-Cys(Acm)^3^-Lys(Boc)^4^-Cys(Trt)^5^-Phe^6^-Phe^7^-DTrp(Boc)^8^-Lys(Boc)^9^-Thr(*^t^*Bu)^10^-Phe^11^-Cys(Trt)^12^-Ser(*^t^*Bu)^13^-Cys(Acm)^14^-O-Cl-Trt-R (AT6S), was washed, dried, and divided in two equal parts. The first was stored at −20 °C and the second was used to complete the synthesis. The protected DOTA-precursor, DOTA(tris(*^t^*Bu)ester) (0.15 mmol, 3 eq), was coupled using HOBt (35 mg, 0.23 mmol) and DIC (25.6 µL, 0.17 mmol) in N,N-dimethylformamide (DMF). The resin was washed and dried under vacuum. The peptide-conjugates were released from the solid support and the protecting groups, except for the Acm, were removed by 4 h treatment with a TFA/triisopropylsilane, (TIS)/ethanedithiol, (EDT)/anisole/H_2_O 93:3:2:1 (*v*/*v*/*v/v*) mixture at room temperature.

#### 4.2.2. Cyclization and Isolation of DOTA-Conjugates

For the formation of the first disulfide bridge, Cys^6^-Cys^11^ (AT5S) or Cys^5^-Cys^12^ (AT6S), the linear peptide was dissolved in 0.01 M phosphate buffer, pH 7.5 (final concentration 1 mg/mL) and 1% DMSO was added. The mixture was stirred at room temperature and the formation of the monocyclic product was monitored by HPLC and MS. For the formation of the second disulfide bridge (Cys^3^-Cys^14^, 12 member ring), oxidation with iodine in a AcOH/H_2_O 4:1 (*v*/*v*) solution after in situ removal of the Acm groups was pursued, as previously described [38]. The bicyclic analogs were isolated by semi-preparative HPLC and lyophilized. Analytical HPLC confirmed the high purity of peptide conjugates and ESI-MS spectra were consistent with the expected formula (Table 1 and Appendix A).

### 4.3. Radiolabeling with In-111 and Quality Control

Labeling of the peptide-conjugates with In-111 was conducted by adding 10 nmol peptide analog per 37 to 74 MBq of [^111^In]InCl_3_ in 0.1 M sodium acetate buffer and 10 mM sodium ascorbate. The typical end pH was 4.6. Labeling was completed by incubation in a boiling water bath for 20 min. Prior to HPLC quality control, EDTA in 0.1 M acetate buffer was added to a final concentration of 1 mM to the labeling reaction mixture as a “free” [^111^In]In^3+^ scavenger. Analyses were performed on a RP18 column using system 2.

### 4.4. In Vitro Studies

#### 4.4.1. Cell Lines and Cell Culture

For in vitro assays, CHO cells transfected to stably express the hSST_1_R were kindly offered by Drs. T. Reisine and G. Singh (University of Pennsylvania, Philadelphia, PA, USA), whereas the CCL39 cells transfected to stably express one of the hSST_2_R, hSST_3_R, or hSST_4_R were a kind gift of Dr. D. Hoyer (Novartis Pharma, Basel, Switzerland). Furthermore, HEK293 cells, transfected to stably express one of the expressing the T7-epitope-tagged hSST_2_R, hSST_3_R or hSST_5_R, were given by Dr. S. Schultz (Institute of Pharmacology and Toxicology, Friedrich Schiller University of Iena, Jena, Germany). CHO cells were cultured in Ham’s F-12 medium, CCL39 cells in a mixture of Dulbecco’s modified Eagle’s medium (DMEM) and Ham’s F-12 medium in a 1:1 *v*/*v* ratio. HEK293 cells were grown in DMEM GLUTAMAX-I. Media were supplemented with 10% fetal bovine serum (FBS), 100 U/mL penicillin and 100 µg/mL streptomycin along with 400–500 µg/mL G418-sulfate. The rat pancreatic tumor cell line AR4-2J, endogenously expressing the SST_2_R, was provided by Prof. S. Mather (St. Bartholomew’s Hospital, London, UK) and cultured in F-12 K medium supplemented as above, but without addition of G418. All culture reagents were obtained from Gibco BRL, Life Technologies (Grand Island, NY, USA) or from Biochrom KG Seromed (Berlin, Germany). Cells were cultured at 37 °C and a humidified 5% CO_2_ atmosphere and weekly passages were performed using a Trypsin/EDTA (0.05%/0.02% *w*/*v*) solution.

#### 4.4.2. Receptor Autoradiography

The hSST_1–5_R affinity profiles of monocyclic AT2S and of the two bicyclic AT5S and AT6S analogs were determined by receptor autoradiography [43,47]. Cell membrane pellets were prepared from CHO-hSST_1_R cells, CCL39-hSST_2_R, CCL39-hSST_3_R, and CCL39-hSST_4_R cells, and HEK293-hSST_2_R cells and stored at −80 °C. Receptor autoradiography was performed on 20-μm-thick cryostat (Microm HM 500, Walldorf, Germany) sections of the membrane pellets mounted on microscope slides, and then stored at −20 °C, as previously described [33]. For each of the tested compounds, complete displacement experiments were performed with the universal SS28 radioligand [^125^I][Leu^8^,DTrp^22^,I-Tyr^25^]SS28 ([^125^I]I-[LTT]SS28) (74 GBq/mmol; Anawa, Wangen, Switzerland) using 15,000 cpm/100 μL and increasing concentrations of the unlabeled peptide ranging from 0.1 to 1000 nM. Native SS14 was run in parallel as an internal control using the same increasing concentrations. The sections were incubated with [^125^I]I-[LTT]SS28 for 2 h at room temperature in 170 mmol/L Tris-HCl buffer (pH 8.2), containing 1% bovine serum albumin (BSA), 40 mg/L bacitracin, and 10 mmol/L MgCl_2_ to inhibit endogenous proteases. The incubated sections were washed twice for 5 min in cold 170 mmol/L Tris-HCl (pH 8.2) containing 0.25% BSA. After a brief dip in 170 mmol/L Tris-HCl (pH 8.2), the sections were dried quickly and exposed for 1 week to Kodak BioMax MR film (Rochester, NY, USA). IC_50_ values were calculated after quantification of the data using a computer-assisted image processing system as described previously [33]. Tissue standards (Autoradiographic [^125^I] and/or [^14^C] microscales, GE Healthcare; Little Chalfont, UK) that contain known amounts of isotope, cross-calibrated to tissue-equivalent ligand concentrations, were used for quantification.

#### 4.4.3. Immunofluorescence Microscopy-Based Internalization Assay

An immunofluorescence microscopy-based internalization assay for hSST_2_R was performed in HEK293-hSST_2_R cells, grown on poly-DLys (20 μg/mL) (Sigma-Aldrich, St. Louis, MO, USA) coated 35-mm four-well plates (Cellstar, Greiner Bio-One GmbH, Frickenhausen, Germany) [44,48]. Cells were treated for 30 min at 37 °C in growth medium containing either the compounds to be tested (AT5S: 1 µM, 10 µM or 10 µM + 10 nM SS14; AT6S: 1 µM) or TOC (10 nM and 1 µM) and SS14 (10 nM), serving as positive controls. HEK293-hSST_2_R cells treated with vehicle alone were used as a negative control. The cells were then rinsed twice with PS (100 mM phosphate buffer containing 0.15 M sucrose), fixed and permeabilized for 7 min with cold methanol (−20 °C), rinsed twice with PS, and then blocked for 60 min at room temperature with PS containing 0.1% BSA. Next, cells were incubated for 60 min at room temperature with the hSST_2_R-specific primary antibody R2-88 (provided by Dr. A. Schönbrunn, Houston, TX, USA) diluted 1:1000 in PS and washed 3 × 5 min with PS containing 0.1% BSA. The cells were incubated for 60 min at room temperature in the dark with the secondary antibody Alexa Fluor 488 goat anti-rabbit IgG (H+ L) (Molecular Probes, Inc., Eugene, OR, USA) diluted in PS (1:600), washed 3 × 5 min with PS containing 0.1% BSA, embedded with PS/glycerol 1:1, and covered with a glass cover slip. The cells were imaged using a Leica DM RB immunofluorescence microscope (Leica, Deerfield, IL, USA) and an Olympus DP10 camera (Olympus Corporation, Shinjuku, Tokyo, Japan).

#### 4.4.4. Radioligand Internalization in AR4-2J and HEK293-hSST_3_R Cells

Radioligand internalization experiments were performed using the rSST_2_R-positive AR4-2J cells and HEK293-hSST_3_R cells [32]. Cells were grown to confluence in six-well plates 24 h prior to the experiment. The following day, the cells were washed twice with ice-cold internalization medium (F-12 K with 1% FBS for the AR4-2J cells and DMEM GLUTAMAX-I with 1% FBS for the HEK293-hSST_3_R cells). They were supplied with fresh medium (1.2 mL) and [^111^In]In-AT6S (150 μL, ≈300,000 cpm/2 pmol peptide) were added per well, followed by 0.5% BSA phosphate buffered saline (PBS) alone (150 μL, total series) or by a 1 μM TATE (AR4-2J cells) or a KE108 (HEK293-hSST_3_R cells) solution (0.5% BSA-PBS; 150 μL, nonspecific series). Cells were incubated at 37 °C for 60 min and incubation was interrupted by the removal of the medium and rapid rinsing with ice-cold 0.5% BSA-PBS. Cells were incubated (2 × 5 min) at ambient temperature in acid wash buffer (50 mM glycine in 0.1 M NaCl, pH 2.8). The supernatants were collected (membrane-bound radioligand fraction). The cells were rinsed with 0.5% BSA-PBS, lysed with 1 N NaOH and collected (internalized radioligand fraction). Collected fractions were measured for their radioactivity content in the γ-counter and the percentage of internalized activity was calculated versus the total added activity per well. Experiments were performed twice in triplicate.

### 4.5. Animal Studies

#### 4.5.1. Metabolic Stability of [^111^In]In-AT5S and [^111^In]In-AT6S in Mice

The radioligand, [^111^In]In-AT5S or [^111^In]In-AT6S, was injected as a 150 μL bolus (11–22 MBq, 3 nmol total peptide) in the tail vein of male Swiss albino mice (30 ± 5 g, NCSR “Demokritos” Animal House, Athens, Greece) [32,36]. At 5 min pi blood (~1mL) was collected from the heart of mice in a prechilled insulin syringe and swiftly placed in an ice-cold polypropylene tube containing EDTA on ice. Blood samples were centrifuged at 2000× *g* at 4 °C for 10 min. The supernatant (>90% radioactivity recovered) was collected and an equal volume of MeCN was added. The mixture was centrifuged for 10 min at 15,000× *g* at 4 °C. The supernatant (>90% recovery of radioactivity) was collected, and the organic solvent was removed under a mild N_2_-stream; the residue was redissolved in physiological saline, passed through a 0.22-μm Millex-GV filter (Millipore, Milford, CT, USA) (>90% recovery of radioactivity), and analyzed by RP-HPLC (>96% radioactivity recovered). An RP18 Symmetry Shield cartridge column (5 μm, 3.9 mm × 20 mm, Waters, Vienna, Austria) was eluted at a 1 mL/min flow rate with the following gradient: 100%A/0%B to 60%A/40%B in 40 min, whereby A = 0.1% aqueous TFA and B = MeCN (system 3). Co-injection of blood samples with aliquots of the original labeling reaction on the HPLC column helped to identify intact [^111^In]In-AT5S or [^111^In]In-AT6S by co-elution in this system (identical *t*_R_).

#### 4.5.2. Biodistribution of [^111^In]In-AT6S in AR4-2J Tumor-Bearing SCID Mice

For biodistribution experiments, male SCID mice of 7 weeks of age (15–20 g) on arrival day (NCSR ‘‘Demokritos’’ Animal House, Athens, Greece) were subcutaneously injected in their flanks with a suspension of AR4-2J cells (150 μL inocula of ~1 × 10^7^ cells in PBS) [32]. The animals were kept under aseptic conditions for 12 days until well-palpable tumors were grown at the inoculation site. On the day of the biodistribution experiment, animals were injected in the tail vein in groups of four with [^111^In]In-AT6S (100 μL, 37 to 74 kBq, 10 pmol total peptide) and were sacrificed at 4 and 24 h pi. In an additional group, animals were co-injected with excess TATE (100 nmol) together with the radioligand (blocked animals) and were sacrificed at 4 h pi. Blood was immediately collected, and the organs of interest were excised and weighed. Sample radioactivity was measured in the γ-counter using proper standards of the injected activity. Data were calculated as %IA/g with the aid of standard solutions and represent mean values ± sd. For comparisons, a Two-way ANOVA with Tukey’s post hoc analysis was applied (PRISM^TM^ GraphPad—6 Software, San Diego, CA, USA). *p* values < 0.05 were considered statistically significant.

#### 4.5.3. Biodistribution of [^111^In]In-AT6S in HEK293-hSST_3_R Tumor-Bearing SCID Mice

For the second biodistribution experiments, male SCID mice of 7 weeks of age (15–20 g) on arrival day (NCSR ‘‘Demokritos’’ Animal House, Athens, Greece) were subcutaneously injected in their flanks with a suspension of HEK293-hSST_3_R cells (150 μL inocula of ~3 × 10^7^ cells in PBS) [32]. The animals were kept under aseptic conditions for 3–4 weeks until well-palpable tumors were grown at the inoculation site. On the day of the biodistribution experiment, animals were injected in the tail vein in groups of four with [^111^In]In-AT6S (100 μL, 37 to 74 kBq, 10 pmol total peptide) alone, or co-injected with excess KE108 (80 nmol) (blocked animals). Mice were sacrificed at 4 h pi, and biodistribution analyses were conducted, as described above.

Mice experiments complied with European and national regulations and study protocols were approved by the Department of Agriculture and Veterinary Service of the Prefecture of Athens (#440448, 01-06-2021 for the stability studies and #440451, 01-06-2021 for the biodistribution and imaging studies).

## 5. Conclusions

This work on bicyclic 6/12-mer and 8/12-mer ring SS14 analogs and their [^111^In]In-labeled versions, has shown that the introduction of two rings was a successful strategy to ensure the metabolic stability of the radioligands in circulation. On the other hand, it revealed that other factors are equally important for effective tumor targeting. Thus, changes in molecule rigidity, ring(s) size and conformation induced by the double ring system directly impacted the SST_1–5_R affinity profile, internalization capacity, and pharmacokinetics. Accordingly, the bicyclic 6/12-mer AT5S/[^111^In]In-AT5S failed to interact with any of the SST_1–5_R, despite its high stability and presence of the “-Phe^7^-DTrp^8^-Lys^9^-Thr^10^-“ pharmacophore sequence. In contrast, the bicyclic 8/12-mer AT6S/[^111^In]In-AT6S behaved like a pansomatostatin and showed superior biological features compared with a previously reported monocyclic 8-mer counterpart (AT4S/[^111^In]In-AT4S). It is interesting to note that unlike other bicyclic analogs with reduced overall number of amino acids, the tetradecapeptide AT6S/[^111^In]In-AT6S retained a pansomatostatin character, albeit showing inferior internalization at the SST_2_R and lower SST_2_R-specific tumor uptake in mice. More studies are warranted to better explore and smartly exploit structural interventions leading to the most advantageous combination of all of the above factors in order to make available true pansomatostatin radiopharmaceuticals for successful application in tumor cancer theranostics.

## Figures and Tables

**Figure 1 ijms-25-01921-f001:**
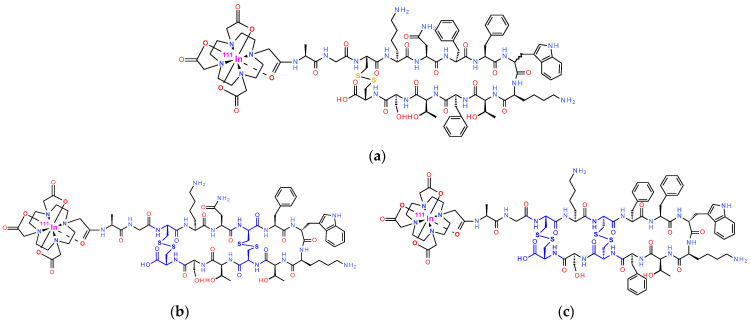
Molecular structures of (**a**) [^111^In]In-AT1/2S: (DOTA-Ala^1^-Gly^2^-c[Cys^3^-Lys^4^-Asn^5^-Phe^6^-Phe^7^-Trp^8^/DTrp^8^-Lys^9^-Thr^10^-Phe^11^-Thr^12^-Ser^13^-Cys^14^-OH], DOTA, 1,4,7,10-tetraazacyclododecane-1,4,7,10-tetraacetic acid), respectively; (**b**) bicyclic 6/12-member-ring [^111^In]In-AT5S (DOTA-Ala^1^-Gly^2^-c[Cys^3^-Lys^4^-Asn^5^-c[Cys^6^-Phe^7^-DTrp^8^-Lys^9^-Thr^10^-Cys^11^]-Thr^12^-Ser^13^-Cys^14^]) and (**c**) bicyclic 8/12-member-ring [^111^In]In-AT6S (DOTA-Ala^1^-Gly^2^-c[Cys^3^-Lys^4^-c[Cys^5^-Phe^6^-Phe^7^-DTrp^8^-Lys^9^-Thr^10^-Phe^11^-Cys^12^]-Ser^13^-Cys^14^]); disulfide bridges in [^111^In]In-AT5S and [^111^In]In-AT6S are highlighted in blue.

**Figure 2 ijms-25-01921-f002:**
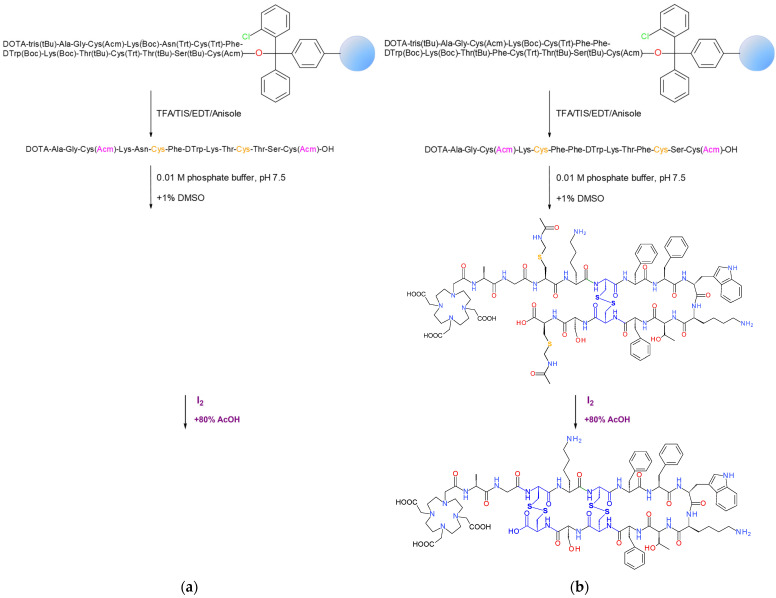
Synthesis of (**a**) bicyclic 6/12-member-ring AT5S and (**b**) bicyclic 8/12-member-ring AT6S, conducted on a solid support. The assembly of the respective protected peptide chains on the resin by Fmoc/*^t^*Bu methodology was followed by the coupling of the protected DOTA-chelator at the N-terminus, release from the resin, and removal of lateral protecting groups by TFA treatment. The formation of the first disulfide bridge (Cys^6^-Cys^11^ for AT5S and Cys^5^-Cys^12^ for AT6S) was achieved with 1% DMSO in phosphate buffer and the second disulfide bridge (Cys^3^-Cys^14^) by I_2_ oxidation in an aqueous acetic acid solution, concomitantly removing the Acm protecting groups.

**Figure 3 ijms-25-01921-f003:**
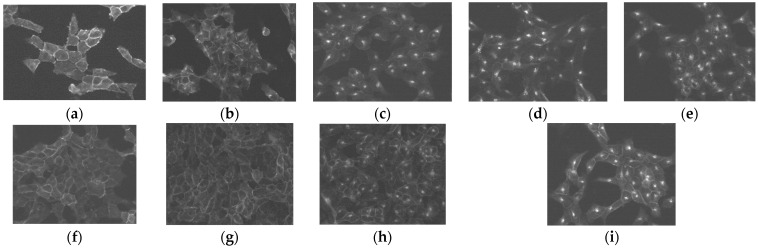
Agonist stimulated hSST_2_R-internalization in HEK293-hSST_2_R cells determined by immunofluorescence microscopy. HEK293-hSST_2_R cells were treated for 30 min at 37 °C with (upper row) (**a**) either vehicle alone (no peptide added, negative control), or (**b**) 10 nM SS14 (positive control), (**c**,**d**) 10 nM and 1 µM TOC, respectively (positive controls), (**e**) 1 µM AT1S (positive control), (lower row) (**f**,**g**) 1 µM and 10 µM AT5S, respectively, (**h**) 10 nM SS14 + 10 µM AT5S, and (**i**) 1 µM AT6S; cells were then processed for immunofluorescence microscopy. Compared to SS14, TOC, and AT1S, AΤ6S is a strong agonist, while AT5S is not an agonist, since it is not able to stimulate hSST_2_R-internalization up to 1 and 10 μM. Moreover, AT5S is not an antagonist either, since it cannot inhibit the SS14-induced internalization of the receptor even in high molar excess over the native hormone (10 µM to 10 nM).

**Figure 4 ijms-25-01921-f004:**
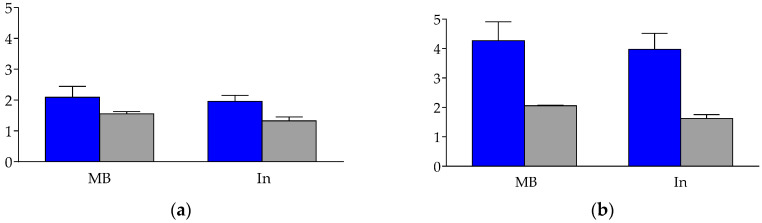
Internalization and cell uptake of bicyclic 8/12-member-ring AT6S [^111^In]In-AT6S in (**a**) AR4-2J cells and (**b**) in HEK293-hSST_3_R cells. Blue bars correspond to the results in the absence of blocker and gray bars to the results in the presence of 1 µM blocker; the blocker is TATE for AR4-2J cells and KE108 for HEK293-hSST_3_R cells. The results represent the mean percentages of total-added activity per well ± sd of at least two experiments performed in triplicate.

**Figure 5 ijms-25-01921-f005:**
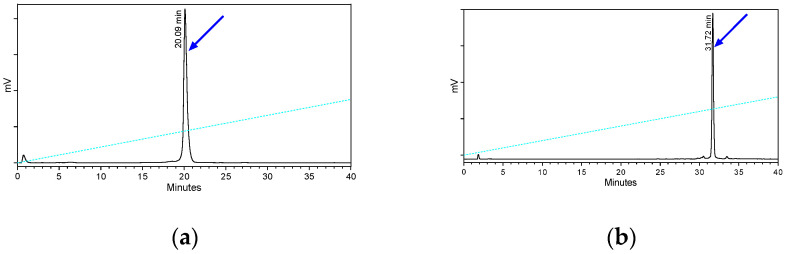
Representative radiochromatograms of blood samples collected 5 min pi of (**a**) bicyclic 6/12-member-ring [^111^In]In-AT5S and (**b**) bicyclic 8/12-member-ring [^111^In]In-AT6S (system 3). The *t*_R_ of the intact radioligands was established after co-injection with radiolabeled samples not administered in mice in the HPLC-column and are indicated with the arrows.

**Figure 6 ijms-25-01921-f006:**
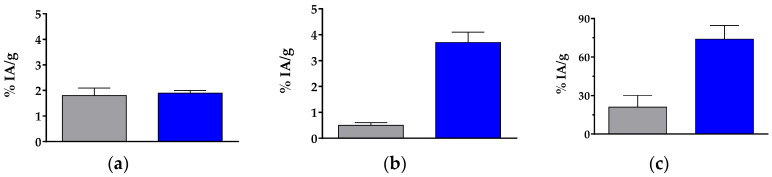
Comparative uptake of [^111^In]In-AT2S (reference [32], gray bar) and [^111^In]In-AT6S (blue bar) in (**a**) AR4-2J tumors, (**b**) HEK293-SST_3_R tumors and (**c**) kidneys in SCID mice at 4 h pi (%IA/g, mean ± sd, n = 4).

**Table 1 ijms-25-01921-t001:** Analytical Data for AT5S and AT6S.

Compound	Rings ^a^	% Purity ^b^	MW Calcd	[M + xH^+^/x] ^c^ Found/Calcd	HPLC *t*_R_ (min), UV Trace	
					System 1 ^d^	System 2 ^e^
AT5S	Cys^6^-Cys^11^(6) Cys^2^-Cys^11^(12)	≥94.0	1934.23	x = 2: 967.3/968.1	8.2	10.34
				x = 3: 645.6/645.7		
				x = 4: 484.5/484.6		
AT6S	Cys^5^-Cys^12^(8) Cys^2^-Cys^11^(12)	≥93.0	2013.37	x = 2: 1007.4/1007.7	30.4	20.06
				x = 3: 672.0/672.1		
				x = 4: 504.7/504.3		

^a^ Cys-residues participating in the ring closures with number of amino acids in the rings in parenthesis. ^b^ Purity was determined by HPLC system 2. ^c^ Fragment ion peaks found by ESI-MS with the calculated mass included. ^d^ System 1: RP-HPLC on an XBridge^TM^ Shield RP18 cartridge column (5 μm, 4.6 mm × 150 mm; Waters, Eschborn, Germany) eluted at a 1 mL/min flow rate with a linear gradient from 80%A/20%B to 60%A/40%B in 40 min; and ^e^ System 2: RP-HPLC on a Symmetry C18 analytical column (3.5 μm, 4.6 mm × 75 mm; Waters, Eschborn, Germany) eluted at a 1 mL/min flow rate with a linear gradient from 90%A/10%B to 50%A/50%B in 20 min, whereby A = 0.1% aqueous TFA, B = MeCN.

**Table 2 ijms-25-01921-t002:** Affinity profile (IC_50_ in nM) of AT5S and AT6S for the five hSST_1–5_R, determined by receptor autoradiography; [^125^I]I-[LTT]SS28 was used as the radioligand and SS14 and AT2S as controls.

Code	hSST_1_R	hSST_2_R	hSST_3_R	hSST_4_R	hSST_5_R
SS14 (12)	1.9 ± 0.5 (5) *	0.7 ± 0.2 (5)	3.3 ± 1.7 (4)	1.6 ± 0.8 (4)	4.2 ± 0.7 (3)
AT2S (12)	14 ± 2 (3)	1.5 ± 0.3 (3)	2.4 ± 0.5 (3)	3.7 ± 0.7 (3)	12 ± 2 (3)
AT5S (6, 12)	>1000 (3)	616 ± 148	>1000 (3)	>1000 (3)	>1000 (3)
AT6S (8, 12)	12 ± 3.3	6.3 ± 0.6	9.7 ± 3.6	5.4 ± 0.8	26 ± 7.0

* Values represent mean IC_50_ values ± SEM in nM with the number of experiments shown in parentheses.

**Table 3 ijms-25-01921-t003:** Biodistribution of [^111^In]In-AT6S in AR4-2J and HEK293-hSST_3_R tumor-bearing SCID mice as %IA/g; values represent mean ± sd (n = 4).

Organs/Tissues	[^111^In]In-AT6S (%IA/g)				
	AR4-2J			HEK293-hSST_3_R	
	4 h Block *	4 h	24 h	4 h Block **	4 h
Blood	0.3 ± 0.08	0.3 ± 0.01	0.07 ± 0.00	0.1± 0.01	0.3 ± 0.02
Liver	17.3 ± 2.8	16.6 ± 1.5	12.1 ± 2.4	10.4 ± 0.4	16.6 ± 1.5
Heart	0.7 ± 0.1	0.5 ± 0.05	0.3 ± 0.06	0.1 ± 0.00	0.5 ± 0.05
Kidneys	49.5 ± 10.5	86.7 ± 10.9	34.1 ± 3.9	26.8 ± 5.4	61.1 ± 10.6
Stomach	0.5 ± 0.2	0.8 ± 0.04	0.3 ± 0.03	0.1 ± 0.04	0.8 ± 0.05
Intestines	2.6 ± 0.6	2.0 ± 0.5	0.6 ± 0.08	0.5 ± 0.1	2.3 ± 0.2
Spleen	6.9 ± 0.9	6.3 ± 1.3	6.0 ± 1.1	4.6 ± 0.3	7.3 ± 1.2
Muscle	0.2 ± 0.04	0.2 ± 0.03	0.1 ± 0.03	0.06 ± 0.01	0.3 ± 0.05
Femur	0.9 ± 0.3	0.9 ± 0.09	0.6 ± 0.1	0.2 ± 0.03	0.8 ± 0.1
Pancreas	0.5 ± 0.1	0.5 ± 0.1	0.4 ± 0.07	0.1 ± 0.00 ***	0.5 ± 0.04
Tumor	0.8 ± 0.05 ***	1.9 ± 0.1	0.8 ± 0.1	0.3 ± 0.05 ***	3.7 ± 0.4

* In vivo SST_2_R blockade by co-injection of 100 nmol TATE; ** In vivo SST_3_R blockade by co-injection of 80 nmol KE108; *** corresponds to *p* < 0.001.

## Data Availability

Data is contained within the article and in the Appendix A.

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
