# Peer review of "Bis(Disulfide)-Bridged Somatostatin-14 Analogs and Their [111In]In-Radioligands: Synthesis and Preclinical Profile"

_ijms, 2024, doi:10.3390/ijms25031921_

Round 1
Reviewer 1 Report
Comments and Suggestions for Authors
In the present manuscript, titled “Bis(Disulfide)-Bridged Somatostatin-14 Analogs and their [111In]In-Radioligands: Synthesis and Preclinical Profile”, Tatsi and co-workers focused on the potential use of somatostatin-like radionuclide carriers for diagnosis and therapy in tumors that overexpress somatostatin receptors (SST1-5R). The aim is to introduce "pansomatostatin" analogs that can broaden clinical indications and enhance the efficacy of current SST2R-preferential radioligands. The findings highlight the need for further research on pansomatostatin-like radioligands for cancer theranostics.
In general, the manuscript is well organized, the methods used are appropriate, and the results are clearly presented and discussed. I think that a summary of the advantages and disadvantages of your method must be reported to improve the easier understanding of the obtained results. This minor edit will enhance the impact and priority of the new method in the field.
Comments on the Quality of English Language
Minor editing of English language required
Reviewer 2 Report
Comments and Suggestions for Authors
The article titled "Bis(Disulfide)-Bridged Somatostatin-14 Analogs and their 2 [111In]In-Radioligands: Synthesis and Preclinical Profile" by Maina and coworkers complies with the standard requirements needed for publication, with some minor changes that will be listed ahead.
- It would be helpful to assign a numbering system to the compounds synthesized and evaluated during this work, making it easier to understand which compound is being referred to in each case and simplifying the text.
- In the results section, the yields obtained for each compound should be included, not just the purity percentage.
- In section "2.3 In vitro studies," it would be convenient to include a better description of the methodology, as well as an explanation of how the IC50 was obtained.
- In section "4.2 Synthesis of the Bicyclic DOTA-SS14 Peptide Conjugates," the full description of the synthesis (methodology, conditions, quantity in grams, mmol, mL, and yields) must be included, along with the complete characterization required by the magazine standards.
- In section "4.4.1. Cell Lines and Cell Culture," a more detailed description of the cells used is required, including characteristics and information on whether they are commercially available, rather than just mentioning that they were gifted.
Reviewer 3 Report
Comments and Suggestions for Authors
In their report, Authors synthesize two bis(disulfide)-bridged bicyclic somatostatin-14 analogs and their [ 111In]In-radioligands based on N-terminally attached DOTA residue. This is a scientifically sound study where the synthesis and radiometal incorporation is followed-up by a pre-clinical profilling. The receptor-binding panel is presented along with the internalization assays in cancer cells, microscopic experiments and the in vivo PK/biodistribution analysis. As a result of these experiments, one of the two bicyclic peptides synthesized has the proper preclinical profile to be useful in imaging tumors over expressing somatostatin receptors 1-5.
I am in favour of publication of this valuable material, I have some minor comments to the Authors:
1) Please consider incorporationd LC/MS chromatograms of compounds and MS spectra as a proof purity and structure
2) [111In]In-AT5S seems not to have a proper receptor profile to be useful for imaging. Therefeore a question arises why the Authors decide to analyze this compound in the single-point in vivo stability assay?
3) I was wondering if the Authors tried to rationalize the lack of binding of AT5S to somatostatin receptors
